# Using a design-based research approach to develop a technology-supported physical education course to increase the physical activity levels of university students: Study protocol paper

**Kuston Sultoni**[1,2]*, **Louisa R. Peralta**[1], **Wayne Cotton**[1]

**1** Sydney School of Education and Social Works, Faculty of Arts and Social Science, The University of Sydney, Sydney, New South Wales, Australia, **2** Faculty of Sport and Health Education, Universitas Pendidikan Indonesia, Bandung, Jawa Barat, Indonesia

* ksul5404@uni.sydney.edu.au

**Data Availability Statement:** No datasets were generated or analysed during the current study. All

## Abstract

### Background

Promoting physical activity (PA) for university students is essential as PA levels decrease during the transition from secondary to higher education. Providing technology-supported university courses targeting students' PA levels may be a viable option to combat the problem. However, it is still unclear how and what technologies should be implemented in university courses to promote PA. This study aims to create a series of design principles for technology-supported physical education courses that aim to increase university students' PA knowledge, motivation and levels.

### Method

The proposed methodology underpinning the research program is a seven-phase design-based research (DBR) approach, with the seven phases encompassed in four sequential studies. These four studies are a systematic review, a qualitative focus group study, a pilot study, and a randomised controlled trial (RCT) study. The protocol paper aims to detail the plan for conducting the four studies in a comprehensive and transparent manner, thus contributing to the methodological evidence base in this field.

### Discussion

Design principles generated from this project will contribute to the growing evidence focusing on effective design and implementation features. Future practitioners can also use these to develop physical education courses that aim to promote university students' physical activity levels, knowledge, and motivation.

relevant data from this study will be made available upon study completion.

**Funding:** The primary author (KS) is supported by the Indonesia Endowment Fund for Education Scholarship/ Lembaga Pengelola Dana Pendidikan Republik Indonesia (LPDP RI) under a doctoral degree scholarship (202001222015860). LPDP RI have no authority in study design; collection, management, analysis, and interpretation of data; writing of the report; and the decision to submit the report for publication.

**Competing interests:** The authors have declared that no competing interests exist.

**Abbreviations:** PA, Physical Activity; DBR, Design-Based Research; PETE, Physical Education Teacher Education; ECTS, European Credit Transfer and Accumulation System; PRISMA, Preferred Reporting Items for Systematic Reviews and Meta-Analyses; PICO, Population, Intervention, Comparator, and Outcome; RCT, Randomized Controlled Trial; LMS, Learning Management System; UX, User Experience; UI, User Interface; IPAQ, International Physical Activity Questionnaire; BREQ-2, Behavioural Regulation in Exercise Questionnaire-2; MET, Metabolic Equivalent of Task.

## Trial registration

The RCT registry number: ACTRN12622000712707, 18/05/2022.

## Background

Interventions focusing on increasing physical activity levels among various age groups, from early childhood to the elderly, have been growing over the last five years [1–4]. These studies have suggested that targeting physical activity at different time points across the lifespan is essential, especially when there is a transition related to educational events [5] (e.g., the transition from preschool to primary school [6–8], from primary to high school [9–12], and from high school to college or university [13, 14]). Yet, the last opportunity to intervene before adulthood is the transition into post-secondary studies. Therefore, this period is ultimately the most important for establishing lifelong actions such as personal, psychosocial, and movement behaviours for those who have not yet established them [15–17]. Physical activity tends to significantly decrease when graduating from high school and enrolling in universities [18] and among first-year university students [15, 19], with 80% of university students not meeting physical activity recommendations during this transition [14]. These studies show that promoting physical activity among university students is essential.

Providing university courses that improve student's physical activity levels may be a viable option, as universities can provide students with access to a range of sports facilities, highly educated facilitators, and appropriate technologies [20]. Furthermore, establishing and maintaining quality physical activity university courses has been a concern. As such, a guideline developed in the US and China [21–23] suggests that administration/support, assessment, instructional strategies, professionalism, learning environment, program staffing, and curriculum are essential facets that promote quality. Previous studies also suggest that providing strategies for administration and directors [24, 25], modelling the development and support for course instructors [26], and utilising technology [27–29] can be viable strategies for increasing the quality of university courses that aim to improve the physical activity levels of university students.

There are various factors that influence physical activity behaviour in young adults [30]. Gaining knowledge of physical activity is considered one of the principle determining factors of physical activity in university-age students [31]. A cross-sectional study involving 258 adults in Hong Kong found that physical activity knowledge had a positive correlation with levels of physical activity, with this correlation strongest among the university student participants [32]. This finding is also supported by a Chinese cross-sectional study recruiting 9826 university students [33], with this study finding that knowledge of the physical activity guidelines was correlated with higher physical activity levels. Thus, physical activity knowledge should be considered as one of the learning outcomes of university courses that promote physical activity in a university setting.

Another important factor that is widely known to be associated with physical activity is motivation [34, 35]. Exercise motivation plays an important role in long-term physical activity behaviour [36]. Systematic reviews examine relationships between motivation and physical activity [37] and examine the effects of physical activity interventions underpinned by motivational principles [38] show that motivation significantly increases physical activity levels [37, 38]. Furthermore, a cross-sectional study involving 1079 participants aged 24±9 years showed that motivation for physical activity and exercise is associated with frequency, intensity, and

duration of exercise [39]. This finding is supported by an observational study using a web-based survey involving 320 wearable activity monitor users, where motivational regulation was correlated with moderate to vigorous physical activity [40]. Hence, having motivational content and outcomes as part of a university course that promotes physical activity in university settings should also be considered.

Research focusing on technologies that promote university students' physical activity levels is gaining more attention. The most common strategy utilised in previous studies has been internet websites [41–49]. Seven out of nine (78%) studies utilising internet websites have been successful in increasing physical activity levels [43–49]. Another common form of technology utilised to enhance university students' physical activity levels are wearable devices. These have ranged from pedometers [50–53] to activity trackers (Misfit, Jawbone UP, Polar M400, Fitbit, and MyWellness Key) [54–58]. Two of four (50%) studies that utilised a pedometer successfully increased participants mean steps per day [53, 59]. Two of five (40%) studies using activity trackers also increased student's physical activity levels [55, 58]. Social media, smartphone applications or mobile apps have also become prominent as a technology to enhance the physical activity levels of university students [60–66]. Three of seven (43%) studies using mobile phone apps have successfully increased students' physical activity levels [62, 63, 65]. However, it is important to highlight that most of the studies (19 of 25 (76%)) utilising technology (internet website, wearable device, mobile phone apps) that aim to increase student's physical activity levels in university settings were non-course-based interventions. That means only six of 25 (24%) studies were course-based interventions, with four of these six studies significantly increasing students' physical activity levels. Nevertheless, it is still unclear what design principles and features of technology are the most effective to support the implementation of these course-based interventions to increase students' physical activity knowledge, motivation and levels. There is also far too little attention paid to creating a set of design principles that can direct future improvement endeavours in this space [67]. Hence, research that focuses on development goals is needed to incorporate technology into courses at universities that aim to increase students' physical activity knowledge, motivation and levels.

The aim of this design-based research project is to create a series of design principles, develop an intervention based on the design principles and measure the impact of the intervention [67–69]. The impact of the intervention will be measured through an increase in physical activity levels, knowledge, and motivation. The purpose of this protocol paper is to fully detail the plan for conducting a design-based research approach to develop a technology-supported physical activity course for increasing university students' physical activity knowledge, motivation and levels in a comprehensive, transparent manner, contributing to the methodological evidence base in this field.

## Aims and objectives

The primary purpose of this research project is to create a series of design principles which can guide the development of a technology-supported physical activity course to increase university student's physical activity knowledge, motivation and levels.

## Methods

### Study design

The research methodology for the study was guided by the Design-Based Research model [67], which simplifies collaboration with practitioners by integrating known and hypothetical design principles with technological affordances to render plausible solutions to these problems. Design-based research was heralded as a practical research methodology that could

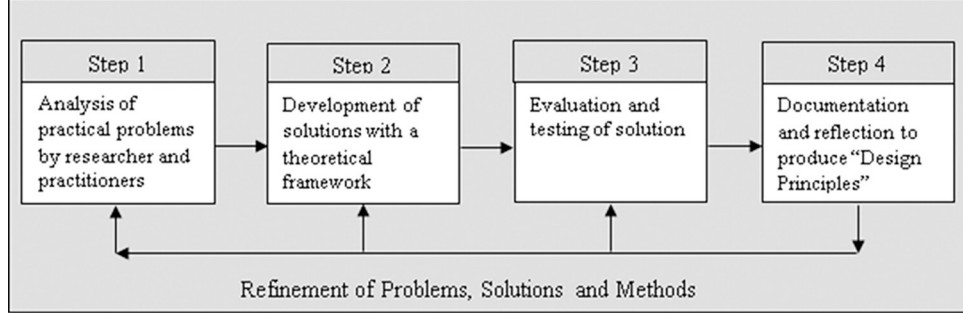

**Fig 1. Reeves' original design-based research model (2000).**

effectively bridge the chasm between research and practice in formal education [70], including physical education [71]. Design-based research can also be used as an alternate model for enquiry in the field of educational technology to make future progress in improving teaching and learning through technology [72]. The original Reeves design-based research model contains four steps (see Fig 1). However, in this study, the four steps will be represented in seven phases to accommodate the cyclic nature of design-based research (see Fig 2). This adaptation has been made because the nature of design-based research is the flexibility of the process that still have some principles [73]. McKenney and Reeves [73] have highlighted that design-based research use: 1) scientific knowledge to ground design work; 2) produce scientific knowledge;

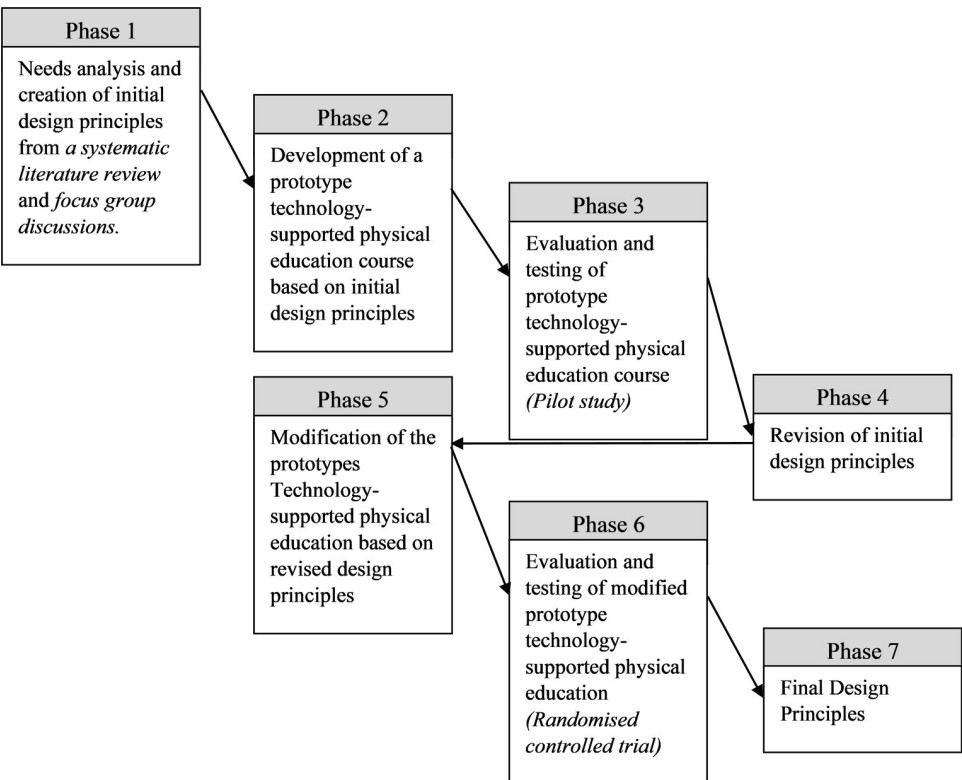

**Fig 2. Diagrammatic overview of how design-based research methodology informed this research project.** The four studies are identified in italics.

3) three main phases (analysis, design, and evaluation phase); and 4) development of both interventions in practise and reusable knowledge. Therefore, the seven phases in this study are: 1) needs analysis and creation of initial design principles from a systematic literature review and focus group discussions; 2) development of a prototype technology-supported physical education course based on initial design principles; 3) evaluation and testing of prototype technology-supported physical education course (pilot study); 4) revision of initial design principles; 5) modification of the prototype technology-supported physical education course based on revised design principles; 6) evaluation and testing of modified prototype technology-based physical education course (randomised controlled trial); and 7) the final design principles. These seven phases will be housed in four sequential studies. These are a: 1) systematic review; 2) qualitative focus group study; 3) pilot study, and 4) randomised controlled trial study. This project has obtained ethics approval from the University of Sydney Human Research Ethics Committee (Project No. 2021/071; Project No. 2021/935). Written informed consent was obtained from all participants before the study commenced.

## Location and setting

This study will be conducted at the Universitas Pendidikan Indonesia (Indonesia University of Education) located in Bandung, West Java, Indonesia (www.upi.edu). The physical activity education course of interest is offered to first year and second-year undergraduate students as required (mandatory) learning or as an elective. To clarify, there is also university courses called Physical Education Teacher Education (PETE) programs that aim to educate preservice physical education teachers to prepare them to become beginning teachers in school settings. However, the physical activity education course (herein now called physical education course) in this study is unit of study with the main outcome is to promote active lifestyles or lifelong physical activity for all undergraduate students. Based on the higher education system in Indonesia, the workload used in the physical education course is a semester credit unit. In this system, one credit is equivalent to 48 hours/semester, 16 meetings, and 3 hours for each meeting consisting of scheduled lecture activities, structured, and independent assignments. The credit system is different from European Credit Transfer and Accumulation System (ECTS) which has a workload of 25 hours of study or 2.5 hours for 10 meetings per semester. When the credit system is changed to the ECTS, the students' workloads for this course is 48:25 x 2 credits = 3.84 ECTS.

## Phase 1. Needs analysis and creation of initial design principles

The purpose of Phase 1 is the creation of the initial design principles from the systematic literature review and needs analysis with key stakeholders. In addition, this phase will also identify appropriate technologies that will facilitate the implementation of a physical education course for increasing university students' physical activity knowledge, motivation and levels.

## Systematic review (study I)

*Purpose*. This systematic review aims to form initial design principles focusing on implementing a physical education course for university settings to improve university students' physical activity knowledge, motivation and levels.

The first study in this project is systematic literature review aims to identify, critically appraise, and summarise the best available evidence regarding the effectiveness of technology-supported university courses for increasing the physical activity knowledge, motivation and levels of university students. This systematic review will follow the Preferred Reporting Items for Systematic Reviews and Meta-Analyses (PRISMA) guidelines [74]. The protocol of the

study will be registered with PROSPERO (ID: CRD42020210327). The process will involve planning a review, searching, and selecting studies, data collection, risk of bias assessment, analysis, and interpreting results. The PICO (Population, Intervention, Comparator, and Outcome) formula is used in this study to limit the question. The population is a university student or college students, or students who enrol in higher institution. Intervention is technology-based physical education or course-based intervention using technology. Technology is limited to online delivery, learning management system, website, wearable device, mobile application, activity tracker, and blended learning using technology. Comparation is studies with a control group including RCT or Non-RCT or quasi-experimental study describing interventions using technology-supported university courses targeting physical activity among university students. The primary outcome will be physical activity levels. Eight electronic bibliographic databases will be sought including CINAHL, ERIC, MEDLINE, ProQuest, PsycINFO, Scopus, SPORTDiscus, and Web of Science from 1st January 2010 to 31st December 2020. 10 item quality assessment scale derived from Van Sluijs and colleagues [75] will be used to measure the quality of selected studies.

## Qualitative focus group study (study 2)

*Purpose*. The purpose of this study is to confirm and add to the initial series of design principles to inform the design of Phase 2.

*Setting access and recruitment*. There will be three focus group discussions with key stakeholders. These stakeholders will include administrators and directors, course instructors/lecturers and students. The first discussion is with key stakeholders such as the curriculum development team, course coordinator and the Dean of the Faculty of Sport and Health Education. The discussion will be guided by a form to confirm that the content of the initial design principles is appropriate considering the learning outcomes and current technologies provided and supported by the university. The second focus group discussion is with lecturers who teach the physical education course. This discussion will be guided with lecturers being asked to answer questions focusing on the system and content of the course. The third focus group discussion will be with students who have previously enrolled in and completed the physical education course with questions focusing on the best technologies for their learning. The semi-structured focus group discussion guidelines are attached as supporting information file [see S1 Appendix].

Participation in these focus groups is voluntary. All participants will be invited using email and will be given the Participant Information Statements. A signed Participant Consent Form will be needed to participate in the focus group discussion.

*Sampling*. There will be three focus group discussions. In the first focus group discussion, four key stakeholders or policy makers will be invited to participate: 1) a physical education course lecturer; 2) the course coordinator; 3) a member of university curriculum development team; and 4) the Dean of the Faculty of Sport and Health Education. The second focus group discussion will involve six randomly selected lecturers from 30 lecturers who currently implement the physical education course. The third focus group discussion will involve 12 undergraduate students who have enrolled in the physical education course previously. The size of sample in the focus group discussion 1, 2 and 3 will enable data saturation.

*Data analysis*. Qualitative data from the focus group discussions will be analysed using the thematic analysis [76]. The analysis will involve transcribing the data before coding individual comments into categories determined by the research question. Each category will then be sub-coded and investigated in more detail. This method will enable issues and themes in the data to emerge and from these issues and themes, conclusions will be able to be made to reinforce the initial design principles derived from the systematic review.

## Phase 2. Development of a prototype technology-supported physical education course based on initial design principles

The purpose of Phase 2 is to develop a prototype based on the initial design principles from Phase 1. The design includes content and technical development. Content design will be reviewed by three physical education experts. Content development will produce a paper-based prototype of the technology that will show how the technology aligned with the initial design principles. While technical development will produce the design of a Learning Management System (LMS) and mobile application (App). Technical design includes User Experience (UX) represented by the flow chart of how LMS and App work and User Interface (UI) represented by a series of visual design on how LMS and App looks like. The next step is producing the LMS, and App based on technical development by IT developers to create a prototype for the technology-based physical education course.

## Phase 3. Evaluation and testing of prototype technology-supported physical education course

The purpose of Phase 3 is to test and evaluate the prototype from Phase 2. In this phase, a pilot study will be conducted to implement and test the prototype from the previous phases to examine the feasibility and acceptability of a randomised controlled trial that will be conducted in Phase 6.

**Pilot study (study 3).** *Purpose*. The pilot study aims to test the prototype with a small sample size to examine the feasibility and acceptability of the intervention.

*Setting access and recruitment*. The pilot study will include two classes of a semester physical education course. One of the classes will be the intervention group, and the other will be the control group. The intervention group will receive a course incorporated with the prototype for the technology-supported physical education course. The control group will receive the usual physical education course. Three outcomes will be measured pre- and post-intervention. This phase also includes developing the research instruments to determine validity and reliability.

There are two recruitment processes in this pilot study including the lecturer and students. In the lecturer recruitment step, the researcher will invite a lecturer who participated in the focus group discussion in Phase 1. The lecturer will be included in the pilot study if he/she meets the following inclusion criteria: 1) run two classes of the physical education course; 2) willing to take part in lecturer training before the physical education course class begins; and 3) willing to invite their students to take part in the pilot study.

The lecturer will invite their students to participate in the pilot study in the first week of the physical education course. Participation in this pilot study is voluntary, and all participants will be given Participant Information Statements and invited to sign a Participant Consent Form to participate. Participation or non-participation in this study will not affect students' scores in physical education courses. The number of students in each class varies from 20–50 students. Students will be included in the pilot study if they meet the following inclusion criteria: 1) students enrolled in a physical education course with a lecturer who will participate in the pilot study; 2) voluntarily participate in the pilot study; and 3) owns an android smartphone.

*Intervention*. The intervention group will receive 16 weeks of the physical education course incorporated with the prototype of technology-supported physical education course implemented by their lecturer. The intervention and control group will have the same content (See Table 1); however, the intervention group will have access to the prototype of technology. The prototype will be developed in Phase 2 using the initial design principles generated from the focus group discussions and systematic review in Phase 1.

**Table 1. Course content for intervention and control group.**

| Week | Intervention Group | Control Group |
|------|-------------------|---------------|
| 1 | Course Introduction | Course Introduction |
| 2 | Physical activity, Healthy and active lifestyle knowledge | Physical activity, Healthy and active lifestyle knowledge |
| 3 | Health-related physical fitness knowledge | Health-related physical fitness knowledge |
| 4 | Goal setting in Physical activity and fitness | Goal setting in Physical activity and fitness |
| 5 | Body composition | Body composition |
| 6 | Aerobic fitness | Aerobic fitness |
| 7 | Muscular fitness and flexibility | Muscular fitness and flexibility |
| 8 | Midsemester exam | Midsemester exam |
| 9 | Nutrition: Healthy eating | Nutrition: Healthy eating |
| 10 | Aerobic activity- aerobic dance | Aerobic activity- aerobic dance |
| 11 | Aerobic activity- GPS-Based activity | Aerobic activity- GPS-Based activity |
| 12 | Muscular strengthening | Muscular strengthening |
| 13 | Flexibility training | Flexibility training |
| 14 | Create an individual fitness training | Create an individual fitness training |
| 15 | Overcome physical activity barrier | Overcome physical activity barrier |
| 16 | Final exam | Final exam |

*Instruments*. Data will be collected on student's physical activity knowledge, motivation and levels, since the outcome of the course is changing student physical activity behaviour, knowledge and motivation to encourage long-life physical activity. Students' physical activity level will be measured by using International Physical Activity Questionnaire (IPAQ) short form. Students' physical activity knowledge will be measured by a questionnaire that will be developed in the pilot study based on literature [32, 33, 77]. A physical education expert will confirm content validity of the questionnaire and a reliability test of the instrument will be conducted in the pilot study. Students' motivation for physical activity and exercise will be measured using The Behavioural Regulation in Exercise Questionnaire-2 (BREQ-2) [78]. The BREQ-2 consists of a 19-item questionnaire of 5 subscales (amotivation, external regulation, introjected regulation, identified regulation, and intrinsic regulation) to assess motivation to exercise. Construct validity studies showed that the BREQ-2 is valid for measuring exercise motivation among university students [79]. Validity and reliability of an Indonesian version of the BREQ-2 will be conducted in this pilot study.

*Statistics and data analysis*. Data will also be gathered from class observations and interviews with the lecturers and students. The class observation will use a semi-structured observation protocol based on ISO 9126 evaluation model for e-learning [80], focusing on functionality, reliability, efficiency, usability, maintainability, and portability of the prototypes that will be analysed for refining design principles.

Descriptive statistics will be presented (mean and standard deviation) for each group separately. Changes in physical activity level (MET: Metabolic Equivalent of Task), Physical activity knowledge and motivation from pre-test and post-test will be assessed using an ANCOVA. As this is a pilot study, the study will be underpowered; therefore, quantitative outcomes will be interpreted only as feasibility and acceptability measures.

## Phase 4. Revise of initial design principles

The purpose of Phase 4 is to modify the initial design principles from Phase 1 based on data gathered from Phase 3 (pilot study, observation, measurement and feedback from lecturer and

students). In this phase, the initial design principles will be revised to guide the prototype for technology-based physical education courses to determine the revised set of design principles.

## Phase 5. Modification of prototype technology-supported physical education based on revised design principles

The purpose of Phase 5 is to modify the prototype based on Phase 4's revised design principles. The revised design principles in the previous phase will be a foundation to redesign and build the technology-based physical education course. The next step is an expert review to find problems and recommendations for the technology. The final step in this phase is the refinement and modification of the technology prototype.

## Phase 6. Evaluation and testing of modified prototype technology-supported physical education (randomised controlled trial)

In this phase, the refined and modified prototype will be tested and evaluated. The randomised controlled trial will be conducted to examine the effectiveness of a technology-supported physical education course on increasing university students' physical activity levels, knowledge, and motivation.

**Randomised controlled trial study (study 4).** This is a two-arm parallel, randomised controlled trial of a technology-supported physical education course intervention for students enrolled in an elective unit of study. The randomised controlled trial protocol is registered at the Australian New Zealand Clinical Trials Registry (ANZCTR). The RCT request number is ACTRN12622000712707, 18/05/2022. The RCT adheres to SPIRIT guidelines for reporting clinical trial study protocols [see S1 Table]. The SPIRIT schedule can be seen in the Fig 3.

*Purpose.* The randomised controlled trial aims to test and evaluate the modified prototype with a larger sample size.

*Hypothesis.* A technology-supported physical education course intervention will increase university student's physical activity levels, knowledge and motivation more effectively than a non-technology-supported physical education courses.

*Setting access and participants.* The randomised controlled trial will include six lecturers and six classes (20 to 50 students each class). The recruitment process and intervention in the randomised controlled trial will be the modified version of the intervention from the pilot study. The six lecturers who agree to participate will be randomly assigned to intervention or control. The lecturer will invite their students to participate in the RCT in the first week of the physical education course. The decision for students who the comparator groups are (the control group) is based on their enrolment teacher. Participation in this randomised controlled trial is voluntary, and all lecturers and students will be given Participant Information Statements and be invited to sign a Participant Consent Form to participate.

*Sample size including power calculation.* The sample size calculation is based on the difference in change in the primary outcome (physical activity) from pre- to post-intervention in both groups (intervention and control group). Based on previous studies of technology-based physical activity interventions among university students, mean effect sizes of around $d = 0.5$ are expected in analyses [47, 81]. To detect such intervention effects in two-sided significance testing ($\alpha = .05$) with a power of 80%, a sample size of 128 participants is required. Considering an expected study drop-out of about 20% and response rates of about 50%, 300 participants will be included in the study.

Number of participants

N = 6 physical education course lecturers

(a) n = 3 will be assigned to the intervention group,

| | STUDY PERIOD | | | | | | | |
| --- | --- | --- | --- | --- | --- | --- | --- | --- |
| | Enrolment | Allocation | Post-allocation | | | | | Close-out |
| TIMEPOINT** | $-t_1$ (August 2022) | 0 | Baseline (September 2022) | $t_2$ | $t_3$ | $t_4$ | etc. | Follow Up (January 2023) |
| **ENROLMENT:** | | | | | | | | |
| **Eligibility screen** | X | | | | | | | |
| **Informed consent** | X | | | | | | | |
| **Teacher training** | | X | | | | | | |
| **Allocation** | | X | | | | | | |
| **INTERVENTIONS:** | | | | | | | | |
| *Technology-supported PE Course* | | | X | | | | | X |
| *PE Course Active Control* | | | X | | | | | X |
| **ASSESSMENTS:** | | | | | | | | |
| *IPAQ-SF* | | | X | | | | | X |
| *BREQ-2* | | | X | | | | | X |
| *Physical activity knowledge quiz* | | | X | | | | | X |
| *Fidelity Check* | | | ←————————————————→ | | | | | |
| *Students FGD* | | | | | | | | X |

**Fig 3. The SPIRIT schedule of enrolment, interventions, and assessments for the technology-supported physical education course.**

(b) n = 3 will be assigned to the control group.
n = 300 students will be randomised 1:1 to either an intervention arm
(a) n = 150 a technology-supported physical education course (intervention group)
(b) n = 150 non-technology-supported physical education course (control group).

*Teacher training.* The lecturers in the intervention group will have teacher training for two days on how to deliver the technology-based physical education course prior to the intervention.

*Intervention.* The intervention group will receive 16 weeks of the physical education course incorporated with modified prototype from the pilot study in Phase 3.

*Instrument.* Data will be gathered using valid and reliable instruments developed from the pilot study and fidelity check and interviews to accommodate the final design principles.

## Phase 7. Final design principles

This is the final phase of the study involving the entire data collected from Phases 1 to 6 to create a final series of design principles to aid future practitioners who will design, implement, and evaluate technology-supported physical education courses for university students.

## Discussion

The purpose of this study is to evaluate a technology-supported physical education course designed to increase the students' physical activity levels, knowledge and motivation. The intervention uses a design-based research approach to determine the technologies and features that are most effective in supporting university students in attaining physical activity and motivational outcomes at one university in Indonesia. Determining the technologies and features will help support and facilitate university students physical activity and motivation, will foster scaling-up and sustainability of this course in this setting and perhaps in other university settings, which is an important facet of publishing protocol papers [82, 83]. This manuscript will report on the systematic creation of a series of design principles using literature, focus group discussions with key stakeholders and modified through cycles of robust research testing. The systematic review study of this project has been completed and published [84]. Four design principles generated from the review will be confirmed and enhanced in the series of focus group discussions with key stakeholders (Phase 1) before the prototype is built from the initial design principles (Phase 2). The prototype will be tested and evaluated in the small sample pilot study to ensure prototype feasibility and validity (Phase 3). Then modifications will be performed based on the pilot study findings (Phase 4) with the modified prototype (Phase 5) tested and evaluated in a large randomised controlled trial study (Phase 6). The final design principles will then be created (Phase 7) to aid future research and practitioners who will design, implement, and evaluate university physical education courses. The design principles generated from this project will contribute to future practitioners designing, implementing, and evaluating technology-supported university physical education courses that aim to enhance university students' physical activity knowledge, motivation and levels. However, this project has limitations. The most important limitation lies in the fact that this project will be conducted during the COVID-19 pandemic, where face-to-face activities in focus groups, as well as the pilot study and randomised controlled trial, will be restricted. In addition, the pandemic will also limit the use of objective measures of physical activity (i.e., accelerometer) as this instrument requires a face-to-face setting and application. Hence, only subjective measures of physical activity levels will be used in this study.

## Supporting information

**S1 Appendix. Semi-structured focus group discussion guidelines.**
(DOCX)

**S1 Table. SPIRIT 2013 checklist: Recommended items to address in a clinical trial protocol and related documents**[*].
(DOCX)

**S1 File.**
(PDF)

## Author Contributions

**Conceptualization:** Kuston Sultoni, Louisa R. Peralta, Wayne Cotton.

**Funding acquisition:** Kuston Sultoni.

**Project administration:** Kuston Sultoni.

**Supervision:** Louisa R. Peralta, Wayne Cotton.

**Writing – original draft:** Kuston Sultoni.

**Writing – review & editing:** Louisa R. Peralta, Wayne Cotton.

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
