## [Decision Letter · Decision Letter 0]

12 Aug 2022

PONE-D-22-15083Using a design-based research approach to develop a technology-supported physical education course to increase the physical activity levels of university students: Study protocol paperPLOS ONE

Dear Dr. Sultoni,

Thank you for submitting your manuscript to PLOS ONE. After careful consideration, we feel that it has merit but does not fully meet PLOS ONE’s publication criteria as it currently stands. Therefore, we invite you to submit a revised version of the manuscript that addresses the points raised during the review process.

We look forward to receiving your revised manuscript.

Kind regards,

Walid Kamal Abdelbasset, Ph.D.

Academic Editor

PLOS ONE

Journal Requirements:

https://journals.plos.org/plosone/s/file?id=ba62/PLOSOne_formatting_sample_title_authors_affiliation.

2. Please expand the acronym “LPDP RI” as indicated in your financial disclosure so that it states the name of your funders in full.

4. We note that the original protocol file you uploaded contains a confidentiality notice indicating that the protocol may not be shared publicly or be published. Please note, however, that the PLOS Editorial Policy requires that the original protocol be published alongside your manuscript in the event of acceptance. Please note that should your paper be accepted, all content including the protocol will be published under the Creative Commons Attribution (CC BY) 4.0 license, which means that it will be freely available online, and any third party is permitted to access, download, copy, distribute, and use these materials in any way, even commercially, with proper attribution.

Therefore, we ask that you please seek permission from the study sponsor or body imposing the restriction on sharing this document to publish this protocol under CC BY 4.0 if your work is accepted. We kindly ask that you upload a formal statement signed by an institutional representative clarifying whether you will be able to comply with this policy. Additionally, please upload a clean copy of the protocol with the confidentiality notice (and any copyrighted institutional logos or signatures) removed.

5. We note that the original protocol that you have uploaded as a Supporting Information file contains an institutional logo. As this logo is likely copyrighted, we ask that you please remove it from this file and upload an updated version upon resubmission.

Additional Editor Comments:

The discussion section needs to be rewritten.

Reviewers' comments:

Reviewer's Responses to Questions

**Comments to the Author**

1. Does the manuscript provide a valid rationale for the proposed study, with clearly identified and justified research questions?

Reviewer #1: Yes

Reviewer #2: Yes

2. Is the protocol technically sound and planned in a manner that will lead to a meaningful outcome and allow testing the stated hypotheses?

Reviewer #1: Yes

Reviewer #2: Yes

3. Is the methodology feasible and described in sufficient detail to allow the work to be replicable?

Reviewer #1: Yes

Reviewer #2: Yes

4. Have the authors described where all data underlying the findings will be made available when the study is complete?

Reviewer #1: Yes

Reviewer #2: Yes

5. Is the manuscript presented in an intelligible fashion and written in standard English?

Reviewer #1: Yes

Reviewer #2: Yes

6. Review Comments to the Author

You may also provide optional suggestions and comments to authors that they might find helpful in planning their study.

Reviewer #1: The subject of this study is judged to be very interesting. Physical activity is an important activity for all of us.

Approaching the seven research programs in this study is judged to be creative. However, it is judged that the discussion part is somewhat weak. Further discussion on the results of this study is needed. It is judged that the quality of the paper will be improved if the discussion of each research stage is added.

On page 9, please correct the notation of prior research in the form.

Reviewer #2: The endeavor of the authors are appreciated, as the topic seems to be relevant and promising. thanks alot for your protocol topic selection

7. PLOS authors have the option to publish the peer review history of their article (what does this mean?). If published, this will include your full peer review and any attached files.

Reviewer #1: No

Reviewer #2: No

---

## [Author Response · Author response to Decision Letter 0]

26 Aug 2022

Reviewer #1: The subject of this study is judged to be very interesting. Physical activity is an important activity 

for all of us.

Approaching the seven research programs in this study is judged to be creative. However, it is judged that the 

discussion part is somewhat weak. Further discussion on the results of this study is needed. It is judged that the 

quality of the paper will be improved if the discussion of each research stage is added.

On page 9, please correct the notation of prior research in the form.

Response:

Thank you for your comment. We have rewritten the discussion section (see Page 12 Line 21-29).

“The systematic review study of this project has been completed and published [85]. Four design 

principles generated from the review will be confirmed and enhanced in the series of focus group 

discussions with key stakeholders (Phase 1) before the prototype is built from the initial design 

principles (Phase 2). The prototype will be tested and evaluated in the small sample pilot study to ensure 

prototype feasibility and validity (Phase 3). Then modifications will be performed based on the pilot 

study findings (Phase 4) with the modified prototype (Phase 5) tested and evaluated in a large 

randomised controlled trial study (Phase 6). The final design principles will then be created (Phase 7) 

to aid future research and practitioners who will design, implement, and evaluate university physical 

education courses.”

Reviewer #2: The endeavor of the authors are appreciated, as the topic seems to be relevant and promising. 

thanks alot for your protocol topic selection

Response:

Thank you for your comment.

---

## [Decision Letter · Decision Letter 1]

22 Sep 2022

PONE-D-22-15083R1Using a design-based research approach to develop a technology-supported physical education course to increase the physical activity levels of university students: Study protocol paperPLOS ONE

Dear Dr. Sultoni,

Thank you for submitting your manuscript to PLOS ONE. After careful consideration, we feel that it has merit but does not fully meet PLOS ONE’s publication criteria as it currently stands. Therefore, we invite you to submit a revised version of the manuscript that addresses the points raised during the review process.

We look forward to receiving your revised manuscript.

Kind regards,

Walid Kamal Abdelbasset, Ph.D.

Academic Editor

PLOS ONE

Reviewers' comments:

Reviewer's Responses to Questions

**Comments to the Author**

1. Does the manuscript provide a valid rationale for the proposed study, with clearly identified and justified research questions?

Reviewer #1: Partly

Reviewer #2: Yes

2. Is the protocol technically sound and planned in a manner that will lead to a meaningful outcome and allow testing the stated hypotheses?

Reviewer #1: No

Reviewer #2: Yes

3. Is the methodology feasible and described in sufficient detail to allow the work to be replicable?

Reviewer #1: Yes

Reviewer #2: Yes

4. Have the authors described where all data underlying the findings will be made available when the study is complete?

Reviewer #1: Yes

Reviewer #2: Yes

5. Is the manuscript presented in an intelligible fashion and written in standard English?

Reviewer #1: Yes

Reviewer #2: Yes

6. Review Comments to the Author

You may also provide optional suggestions and comments to authors that they might find helpful in planning their study.

Reviewer #1: This study is considered very interesting. Physical education in particular seems to be a subject that is being studied in all countries. However, it is found that the reference in the introduction is incorrect.

At the end of the introduction, the word aim and the word purpose are used indiscriminately and redundantly, so they need to be corrected. In addition, the purpose of the study is explained again, and it is judged that the introduction has been sufficiently explained.

The research method was well described. No study results are presented. This is a very dangerous the paper. In the discussion, the results of the study were not compared and analyzed with previous studies. This study lacks the interpretation or argument of the researcher to clarify the necessity.

This study is considered insufficient for publication in the journal plos.

Reviewer #2: The study potocol was well-designed and the statistics are good

7. PLOS authors have the option to publish the peer review history of their article (what does this mean?). If published, this will include your full peer review and any attached files.

Reviewer #1: No

Reviewer #2: No

---

## [Author Response · Author response to Decision Letter 1]

30 Sep 2022

Rebuttal letter

Kuston Sultoni1,2*, Louisa Peralta1 and Wayne Cotton1 

1Sydney School of Education and Social Work, The University of Sydney, 

2Faculty of Sports and Health Education, Universitas Pendidikan Indonesia, 

*ksul5404@uni.sydney.edu.au

September 29, 2022

Dear Editorial Board of PLOS ONE,

We would like to thank the editor and reviewers for the opportunity to revise and resubmit our manuscript PONE-D-22-15083 entitled “Using a design-based research approach to develop a technology-supported physical education course to increase the physical activity levels of university students: Study protocol paper.” 

Please find our responses to the reviewers’ comments as follow:

Reviewer #1: 

1. This study is considered very interesting. Physical education in particular seems to be a subject that is being studied in all countries. 

Response:

Thank you for your comment. 

2. However, it is found that the reference in the introduction is incorrect. 

Response:

Thank you for your comment. We have double-checked the reference list, we did not find the incorrect reference in the introduction.

3. At the end of the introduction, the word aim and the word purpose are used indiscriminately and redundantly, so they need to be corrected. In addition, the purpose of the study is explained again, and it is judged that the introduction has been sufficiently explained.

Response:

Thank you for your feedback on the last paragraph. We have revised the last paragraph to align with the feedback. It now reads:

The aim of this design-based research project is to create a series of design principles, develop an intervention based on the design principles and measure the impact of the intervention [68-70]. The impact of the intervention will be measured through an increase in physical activity levels, knowledge, and motivation. The purpose of this protocol paper is to fully detail the plan for conducting a design-based research approach to develop a technology-supported physical activity course for increasing university students’ physical activity knowledge, motivation and levels in a comprehensive, transparent manner, contributing to the methodological evidence base in this field. 

4. The research method was well described. 

Response:

Thank you for your comment. 

5. No study results are presented. This is a very dangerous the paper. 

Response:

As this paper is a study protocol paper, we consider it very low risk as no results are presented. Please check protocol paper guidelines on PlosOne: https://journals.plos.org/plosone/s/journal-information#loc-scope

6. In the discussion, the results of the study were not compared and analyzed with previous studies. This study lacks the interpretation or argument of the researcher to clarify the necessity. This study is considered insufficient for publication in the journal plos.

Response:

As this paper is a study protocol paper, therefore no study results are discussed in the discussion section. 

Reviewer #2:

1. The study potocol was well-designed and the statistics are good

Response:

Thank you for your comment. 

Thank you for your consideration of our manuscript.

Sincerely,

Kuston Sultoni

---

## [Decision Letter · Decision Letter 2]

16 Nov 2022

Using a design-based research approach to develop a technology-supported physical education course to increase the physical activity levels of university students: Study protocol paper

PONE-D-22-15083R2<gwmw style="display:none;"></gwmw>

Dear Dr. Sultoni,

We’re pleased to inform you that your manuscript has been judged scientifically suitable for publication and will be formally accepted for publication once it meets all outstanding technical requirements.

Kind regards,

Walid Kamal Abdelbasset, Ph.D.

Academic Editor

PLOS ONE

Additional Editor Comments (optional):

Reviewers' comments:

Reviewer's Responses to Questions

**Comments to the Author**

1. Does the manuscript provide a valid rationale for the proposed study, with clearly identified and justified research questions?

Reviewer #2: Yes

2. Is the protocol technically sound and planned in a manner that will lead to a meaningful outcome and allow testing the stated hypotheses?

Reviewer #2: Yes

3. Is the methodology feasible and described in sufficient detail to allow the work to be replicable?

Reviewer #2: Yes

4. Have the authors described where all data underlying the findings will be made available when the study is complete?

Reviewer #2: Yes

5. Is the manuscript presented in an intelligible fashion and written in standard English?

Reviewer #2: Yes

6. Review Comments to the Author

You may also provide optional suggestions and comments to authors that they might find helpful in planning their study.

Reviewer #2: thanks alot for your fast response

7. PLOS authors have the option to publish the peer review history of their article (what does this mean?). If published, this will include your full peer review and any attached files.

Reviewer #2: No

---

## [Editor Report · Acceptance letter]

22 Nov 2022

PONE-D-22-15083R2 

Using a design-based research approach to develop a technology-supported physical education course to increase the physical activity levels of university students: Study protocol paper 

Dear Dr. Sultoni:

I'm pleased to inform you that your manuscript has been deemed suitable for publication in PLOS ONE. Congratulations! Your manuscript is now with our production department. 

Kind regards, 

on behalf of

Dr. Walid Kamal Abdelbasset 

Academic Editor

PLOS ONE